# Weight Gain after Hormone Receptor-Positive Breast Cancer

Archita Goyal [1], Gabrielle E. Milner [2], Ashley Cimino-Mathews [3], Kala Visvanathan [4], Antonio C. Wolff [4], Dipali Sharma [4] and Jennifer Y. Sheng [4,*]

[1] Department of Pathology, Johns Hopkins University, Baltimore, MD 21218, USA; archita.goyal@tufts.edu

[2] Johns Hopkins University School of Medicine, Baltimore, MD 21205, USA; gmilner2@jhmi.edu

[3] Department of Pathology, Johns Hopkins University School of Medicine, Baltimore, MD 21205, USA; acimino1@jhmi.edu

[4] Department of Oncology, Johns Hopkins University School of Medicine, Baltimore, MD 21205, USA; kvisvan1@jhu.edu (K.V.); awolff@jhmi.edu (A.C.W.); dsharma7@jhmi.edu (D.S.)

* Correspondence: jsheng7@jhmi.edu

**Abstract:** Obesity following breast cancer diagnosis is associated with poor overall survival. Understanding weight trajectories will help inform breast cancer survivors at greater risk of weight gain, and those who would benefit from earlier anti-obesity interventions. We performed a retrospective chart review of women from the Breast Cancer Program Longitudinal Repository (BCPLR) at Johns Hopkins diagnosed with hormone receptor-positive Stage I-III breast cancer from 2010 to 2020. We investigated obesity (measured by body mass index [BMI]) over time, patient and tumor characteristics, as well as treatment and recurrence. We observed a significant $\geq 5\%$ increase in BMI from diagnosis to most recent follow-up ($p = 0.009$), particularly among those who were overweight at diagnosis ($p = 0.003$). Additionally, among those up to 5 years since diagnosis, there was a significant association between experiencing a $\geq 0.1 \text{ kg/m}^2$ increase per year since diagnosis and baseline BMI status ($p = 0.009$). A $\geq 0.6 \text{ kg/m}^2$ decrease in BMI was observed for participants with obesity at diagnosis ($p = 0.006$). Our study highlights (i) the significant burden of obesity in women with a history of breast cancer and (ii) higher risks for increases in BMI and shifts in class of obesity among women who are overweight at diagnosis.

**Keywords:** breast cancer; obesity; weight gain

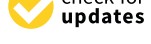



## 1. Introduction

Breast cancer is the most commonly diagnosed cancer among females, with more than 3.8 million individuals with a history of early-stage breast cancer currently living in the United States [1,2]. Approximately 80% of all breast cancer cases are classified as estrogen receptor (ER) positive, and of these 65% are concurrently progesterone receptor (PR) positive [3]. Studies demonstrate that ER-positive and/or PR-positive breast cancers are more likely to recur after five years of diagnosis than during the first five years of treatment [4]. Further research to understand the factors that influence outcomes in this ER-positive and/or PR-positive subtype is critical.

Obesity contributes to risk for both female and male breast cancer, as well as poor outcomes. Obesity is frequently defined using the following subcategories of body mass index (BMI) as defined by the World Health Organization: Class I (30–34.9 $\text{kg/m}^2$), Class II (35–39.9 $\text{kg/m}^2$), and Class III ($\geq 40 \text{ kg/m}^2$). In the United States, obesity prevalence has steadily risen in the last decade with an age-adjusted rate of 42.4% as of 2018 [5]. Women in the highest quintile of BMI are twice as likely to die from breast cancer compared to women without obesity [6]. Endocrine therapy with tamoxifen is associated with weight gain/obesity, with a higher risk for women who first undergo chemotherapy [7]. During menopause, estradiol levels decrease and often contribute to weight gain/obesity as well [8]. Furthermore, rates of recurrence in breast cancer patients with obesity are significantly

higher than those without obesity, and women with obesity and ER-positive breast cancer have increased risk of overall mortality [9,10].

While obesity at diagnosis correlates with poor prognosis, weight gain after diagnosis is also concerning. Individuals who gain 10% or more weight since diagnosis have higher risk of all-cause mortality and breast cancer mortality compared to individuals who maintain (within 5%) or lose weight [11,12]. Both 1-unit and 5-unit increases in BMI have been associated with an increased risk of breast cancer recurrence and death. Other studies have associated adverse outcomes with weight gain of $\geq 0.1$ kg/m$^2$/year within the first five years of diagnosis [13,14].

Race also contributes to risk for worse breast cancer outcomes. While the 5-year survival rate of all women with localized or regional breast cancer is 99% and 86%, respectively, these survival rates are 9% lower in black women compared to white women [15]. Black women have shorter disease-free survival and are 1.22 times more likely to suffer from breast cancer mortality than white women [10,16]. The prevalence of obesity in black women is almost twice that of white women, and there is modest exploration of how both obesity and race impact cancer outcomes [10].

While there is broad research on the influence of obesity on breast cancer mortality, there is limited literature on associations and changes in weight over time among a diverse cohort of breast cancer survivors with hormone receptor (HR)-positive breast cancer. Given that endocrine therapy and postmenopausal state are associated with obesity, it is critical to study HR-positive breast cancer, which incorporates both of these in the treatment plan. Based on these gaps, the primary aims of this study were (1) to assess changes in BMI over time from diagnosis to recent follow up in our patient population of women with early-stage HR-positive breast cancer and (2) to identify any associations between percent and unit increases in BMI and initial BMI status (healthy, overweight, or obese). A secondary objective was to evaluate whether race interacts with BMI status to predict recurrence.

## 2. Materials and Methods

### 2.1. Study Population

Patients included in this analysis were participants in the Johns Hopkins IRB-approved Breast Cancer Program Longitudinal Repository (BCPLR) study (IRB #NA_00019811, PI: Antonio C. Wolff, MD). Key eligibility criteria for the BCPLR included all patients who had received care through the Johns Hopkins Sidney Kimmel Cancer Women's Malignancy Program from 2010 to present. All participants in the BCPLR signed informed consent to allow for future research access to residual specimens and access to patient information available in the electronic medical record. For this study, we identified BCPLR participants who met the following criteria: women, HER-2 negative status, and ER and/or PR positive status. ER or PR status was considered positive if $\geq 1\%$ of tumor cells demonstrated positive nuclear staining after conducting an immunohistochemistry assay [17]. We excluded patients for the following reasons: male, HER-2 positive, ER- and PR-negative, non-invasive breast cancer, rare subtypes other than invasive ductal and invasive lobular cancer (i.e., Paget, phyllodes), metastatic breast cancer, unavailable pathology, unavailable receptor status, or deceased at time of chart review. Of 730 patients in the registry, we reviewed charts for 225. A total of 72 were excluded due to missing BMI at diagnosis and 13 were excluded due to missing treatment information. A total of 140 patients were included in this analysis (Figure 1).

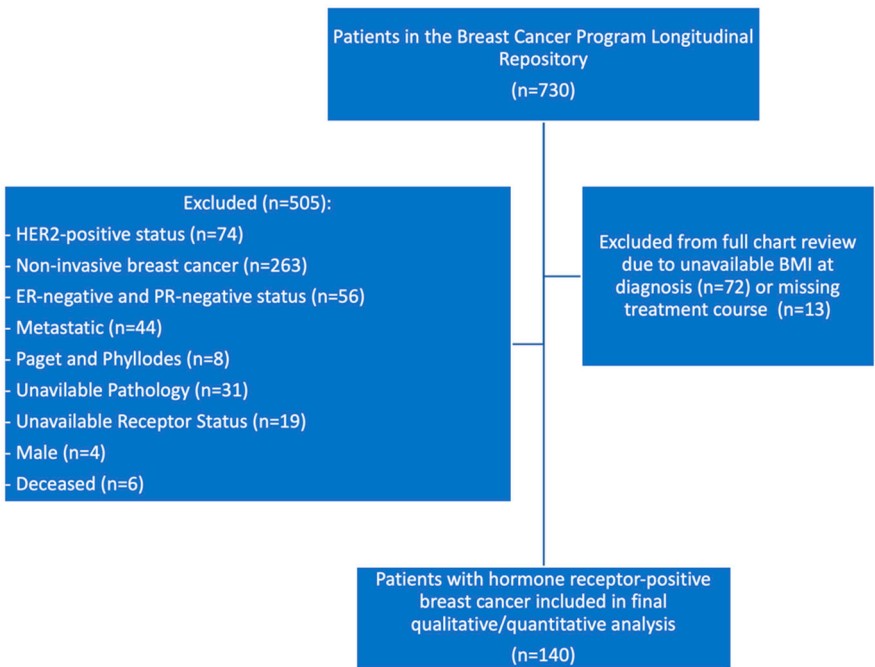

**Figure 1.** Flowchart of included participants.

*2.2. Outcome Measures*

Demographic data from the chart review were collected on: race (Black, Asian, or White), age (years), ethnicity (Hispanic or non-Hispanic), marital status at time of chart review (Divorced/Separated, Married/Partnered, or Single/Widowed), family history of breast cancer in a first degree relative (yes or no), family history of ovarian cancer in a first degree relative (yes or no), tobacco history (never smoker, former smoker, or current smoker), pack years (<20 or ≥20 pack years), alcohol history (yes, no, not currently, or unknown), and drinks per week (<7 or ≥7 drinks per week).

Clinical parameters included: date of diagnosis, time from diagnosis (in months), BMI within three months of diagnosis, date of most recent BMI (as of date of chart review), overall pathologic stage (I, II, or III), receptor status (ER, PR, HER2), pathologic tumor size, surgery type (lumpectomy or mastectomy), multifocal cancer status (yes or no), number of nodes with macro-metastases/micro-metastases/isolated tumor cells, Nottingham histologic grade (1, 2, or 3), histology (ductal, lobular, ductal and lobular, or other), receipt of chemotherapy (yes or no), chemotherapy regimen (TC consisting of Docetaxel and Cyclophosphamide, ddAC-T consisting of dose dense doxorubicin and cyclophosphamide with weekly paclitaxel, or both TC and ddAC-T), receipt of radiation (yes or no), receipt of endocrine therapy (yes or no), metastatic at follow up (yes or no), and recurrence (yes or no).

Weight designation based on BMI classifications by the World Health Organization are as follows: underweight (<18.5 kg/m$^2$), healthy (18.5–24.9 kg/m$^2$), overweight (25–29.9 kg/m$^2$), class I obesity (30–34.9 kg/m$^2$), class II obesity (35–39.9 kg/m$^2$), and class III obesity (≥40 kg/m$^2$). Pathologic tumor size was categorized as follows: T1 (tumor diameter ≤ 2 cm), T2 (tumor diameter = 2.1–5 cm), and T3 (tumor diameter > 5 cm). Most variables were already collected for each patient listed in the registry.

*2.3. Statistical Analysis*

2.3.1. Patient Characteristics by BMI Category

Descriptive statistics were utilized to examine patient, tumor, and treatment characteristics. Participants in the underweight and healthy BMI ranges were collapsed into one category due to the small number of participants in the underweight category (*n* = 2), and

women with class I, II, and III obesity were collapsed into one category, such that the BMI categories were healthy, overweight, and obese.

Categorical variables, such as marital status and race, were described using frequencies with percentages. Continuous variables, such as age and BMI, were described by mean and standard deviation (SD). Frequencies, means, and SD were computed both for the total sample and for each BMI group. Pearson's $\chi^2$ tests compared patient, tumor, and treatment characteristics across BMI groups. Significant associations were further examined using Bonferroni adjusted z-tests for column proportions. Independent-sample *t*-tests assessed any differences between the BMI groups for age at diagnosis. Correlations, independent *t*-tests, and $\chi^2$ tests, respectively, tested whether any continuous and categorical factors were associated with alcohol and tobacco use. Spearman's rank-order correlation in investigations of associations with alcohol and tobacco use measured the monotonic association for non-continuous ranked variables such as cancer stage, and alcohol and smoking category. *T*-test and chi-square tests were utilized as these should be robust to the normality assumption due to sample size.

### 2.3.2. Changes in BMI from Diagnosis to Recent Follow-Up

To evaluate changes in BMI over time, initial and recent BMI scores were utilized to categorize participants according to whether or not they experienced a change in BMI score. Weight outcomes were evaluated by year (i.e., total average increase or decrease per year). Specifically, the prevalence of the following BMI changes were assessed: any increase, $\geq 5\%$ BMI increase, $\geq 0.1$ kg/m$^2$ increase in BMI per year since diagnosis, $\geq 0.6$ kg/m$^2$ increase in BMI, or $\geq 0.6$ kg/m$^2$ decrease in BMI. To identify any association between initial BMI status and subsequent BMI change, $\chi^2$ tests assessed which BMI group may be more likely to experience a specific BMI change. The range of total percent changes in BMI experienced by patients were also computed for each BMI group. Paired *t*-tests assessed the overall mean differences between participants' BMI at diagnosis and at most recent visit for the full sample, and were repeated for each BMI group and again for the participants for whom an increase in BMI was observed.

Changes in BMI were compared based on time since diagnosis. One-way ANOVA with Tukey's post hoc adjustment was used to determine differences in mean BMI and total change in BMI between groups defined by different timepoints since diagnosis (within 36 months of diagnosis, 36–60 months of diagnosis, and greater than 60 months). This was not adjusted for time differences given the likely small number of patients with recurrence. *T*-tests determined such differences between groups defined by <5 years and >5 years since diagnosis.

### 2.3.3. Changes in BMI Classification

To identify trends in shifts in BMI status, the prevalence of patient shifts in BMI category due to BMI changes over time were calculated. Patients were characterized by whether they remained in their initial BMI group, shifted upward to an overweight or obese BMI group, or shifted downward to an overweight or healthy BMI group. Additionally, changes in obesity prevalence among study participants were assessed by two-sided McNemar's tests.

All analyses were conducted using the Statistical Package for Social Sciences (SPSS) version 27 software [18].

## 3. Results

### 3.1. Participant Characteristics

The study population included 140 females with hormone receptor-positive breast cancer with a mean age at diagnosis of 57.2 years (range 30–90) (Table 1). At diagnosis, 28.6% had a healthy or underweight BMI (*n* = 40), 37.9% were overweight (*n* = 53), and 33.6% were obese (*n* = 47). Among the 33.6% of participants with obesity, 49% had class I obesity (BMI 30–34.9), 32% class II obesity (BMI 35–39.9), and 19% class III obesity

(BMI $\geq$ 40). Most participants were white (82%), followed by black (13%), and Asian (5%). Only one participant identified as Hispanic. The majority of participants were married or had a significant other (69%), followed by single/widowed (22%), and divorced/legally separated (9%). A total of 20.7% of all participants had a family history of breast cancer, and of those who were overweight, 28.3% had family history (*n* = 15).

**Table 1.** Characteristics of included women at diagnosis (*n* = 140) from the Breast Cancer Program Longitudinal Repository (BCPLR) at Johns Hopkins University.

| Characteristic | Total | Healthy/Underweight (BMI < 25) | Overweight (25 ≤ BMI < 30) | Obesity (BMI ≥ 30) | *p*-Value |
|---|---|---|---|---|---|
| | N (%) | N (%) | N (%) | N (%) | |
| **BMI Group** | 140 (100) | 40 (28.6) | 53 (37.9) | 47 (33.6) | |
| **Age at Diagnosis (years)** | | | | | |
| Mean (SD) | 57.2 (12.4) | 53.6 (13.1) | 58.9 (12.6) | 58.5 (11.3) | 0.094 |
| ≥50 years | 97 (69.3) | 21 (52.5) | 41 (77.4) | 35 (74.5) | 0.023 |
| **Sex** | | | | | |
| Female | 140 (100) | 40 (100) | 53 (100) | 47 (100) | |
| **Marital Status** | | | | | 0.676 |
| Divorced | 12 (8.6) | 3 (7.5) | 4 (7.5) | 5 (10.6) | |
| Married | 97 (69.3) | 28 (70) | 40 (75.5) | 29 (61.7) | |
| Single | 31 (22.1) | 9 (22.5) | 9 (17.0) | 13 (27.7) | |
| **Race** | | | | | 0.003 |
| White | 115 (82.1) | 31 (77.5) | 47 (88.7) | 37 (78.7) | |
| Black | 18 (12.9) | 3 (7.5) | 5 (9.4) | 10 (21.3) | |
| Asian | 7 (5) | 6 (15) | 1 (1.9) | 0 (0) | |
| **Ethnicity** | | | | | |
| Hispanic | 1 (0.7) | 0 (0) | 0 (0) | 1 (2.1) | 0.373 |
| **Family History** | | | | | |
| Ovarian Cancer | 5 (3.6) | 0 (0) | 2 (3.8) | 3 (6.4) | 0.277 |
| Breast Cancer | 29 (20.7) | 7 (17.5) | 15 (28.3) | 7 (14.9) | 0.214 |
| **Alcohol History** | | | | | 0.425 |
| Yes | 61 (43.6) | 17 (42.5) | 26 (49.1) | 18 (38.3) | |
| No | 75 (53.6) | 22 (55) | 27 (50.9) | 26 (55.3) | |
| Not Currently | 2 (1.4) | 1 (2.5) | 0 (0) | 1 (2.1) | |
| Unknown | 2 (1.4) | 0 (0) | 0 (0) | 2 (4.3) | |
| **Drinks per week** | | | | | 0.778 |
| 0 | 97 (69.3) | 28 (70) | 35 (66) | 34 (72.3) | |
| 1–6 | 36 (25.7) | 9 (22.5) | 15 (28.3) | 12 (25.5) | |
| ≥7 | 7 (5) | 3 (7.5) | 3 (5.7) | 1 (2.1) | |
| **Tobacco History** | | | | | 0.667 |
| Never Smoker | 85 (60.7) | 26 (65) | 33 (62.3) | 26 (55.3) | |
| Former Smoker | 53 (37.9) | 13 (32.5) | 19 (35.8) | 21 (44.7) | |
| Current Smoker | 2 (1.4) | 1 (2.5) | 1 (1.9) | 0 (0) | |
| **Pack Years** | | | | | 0.179 |
| <20 Pack Years | 132 (94.3) | 40 (100) | 49 (92.5) | 43 (91.5) | |
| ≥20 Pack Years | 8 (5.7) | 0 (0) | 4 (7.5) | 4 (8.5) | |

Of all participants, 43.6% had a current history of alcohol, with among those 59% consuming 1–6 drinks per week and 11.5% consuming ≥7 drinks per week. Over one third of all participants (37.9%) were former smokers, and 1.4% were current smokers. Among those who were current or former smokers, 14.5% had ≥20 pack years. A significant relationship was found between history of past or current tobacco use and currently

consuming at least one drink a week ($\chi^2$ (1) = 5.25, $p$ = 0.022). There were no additional significant associations between alcohol consumption or smoking status with BMI group or weight gain. Among those with obesity at diagnosis, 78.7% were white and 21.3% were black. There was a significant relationship between black race and obesity at diagnosis ($\chi^2$(1) = 4.476, $p$ = 0.034). A higher proportion of black participants were obese at diagnosis compared to women of other races (55.6% vs. 30.3%).

### 3.2. Tumor and Treatment Characteristics

About half of participants (59.3%) were stage I, one third (33.6%) were stage II, and 7.1% were stage III (Table 2). Of all pathologic tumor sizes, the majority (71.4%) were classified as T1, followed by 24.3% with T2, and 4.3% with T3. The majority of participants (62.1%) had ductal histology, 19.3% were lobular, and 14.3% were classified as having overlapping features. A quarter of participants had multifocal tumors. Over half of participants (62.1%) received a lumpectomy and one third (37.9%) received a mastectomy. The majority of participants (67.1%) received radiation, reflecting the proportion of patients undergoing breast conserving surgery.

**Table 2.** Tumor and Treatment Characteristics by BMI Status ($N$ = 140).

| Characteristic | Total | Healthy/Underweight (BMI < 25) | Overweight (25 ≤ BMI < 30) | Obesity (BMI ≥ 30) | *p*-Value |
|---|---|---|---|---|---|
| | *N* (%) | *N* (%) | *N* (%) | *N* (%) | |
| **Overall Pathologic Stage** | | | | | 0.828 |
| I | 83 (59.2) | 25 (62.5) | 33 (62.3) | 25 (53.2) | |
| II | 47 (33.6) | 12 (30) | 16 (30.2) | 19 (40.4) | |
| III | 10 (7.1) | 3 (7.5) | 4 (7.5) | 3 (6.4) | |
| **Pathologic Tumor Size** | | | | | 0.991 |
| T1 | 100 (71.4) | 29 (72.5) | 37 (69.8) | 34 (72.3) | |
| T2 | 34 (24.3) | 9 (22.5) | 14 (26.4) | 11 (23.4) | |
| T3 | 6 (4.3) | 2 (5) | 2 (3.8) | 2 (4.3) | |
| **Histologic Grade** | | | | | 0.498 |
| 1 | 28 (20) | 11 (27.5) | 9 (17) | 8 (17) | |
| 2 | 89 (63.6) | 21 (52.5) | 37 (69.8) | 31 (66) | |
| 3 | 23 (16.4) | 8 (20) | 7 (13.2) | 8 (17) | |
| **Multifocal Tumor** | 35 (25) | 7 (17.5) | 16 (30.2) | 12 (25.5) | 0.374 |
| **Histology** | | | | | 0.106 |
| Ductal | 87 (62.1) | 25 (62.5) | 29 (54.7) | 33 (70.2) | |
| Lobular | 27 (19.3) | 9 (22.5) | 15 (28.3) | 3 (6.4) | |
| Mixed | 20 (14.3) | 6 (15) | 6 (11.3) | 8(17) | |
| Other | 6 (4.3) | 0 (0) | 3 (5.7) | 3(6.4) | |
| **Surgery** | | | | | 0.043 |
| Lumpectomy | 87 (62.1) | 22 (55) | 29 (54.7) | 36 (76.6) | |
| Mastectomy | 53 (37.9) | 18 (45) | 24 (45.3) | 11 (23.4) | |
| **Chemotherapy** | 48 (34.3) | 17 (42.5) | 15 (28.3) | 16 (34) | 0.360 |
| TC | 19 (39.6) | 7 (41.2) | 5 (33.3) | 7 (43.8) | |
| ddAC-T | 27 (56.3) | 9 (52.9) | 10 (66.7) | 8 (50) | |
| Both | 2 (4.2) | 1 (5.9) | 0 (0) | 1 (6.3) | |
| **Radiation** | 94 (67.1) | 26 (65) | 34 (64.2) | 34 (72.3) | 0.646 |
| **De Novo Metastatic Disease** | 6 (4.3) | 3 (7.5) | 2 (3.8) | 1 (2.1) | 0.455 |
| **Recurrence** | 10 (7.1) | 5 (12.5) | 3 (5.7) | 2 (4.3) | 0.287 |
| Local | 5 (50) | 2 (40) | 1 (33.3) | 2 (100) | |
| Distant | 5 (50) | 3 (60) | 2 (66.7) | 0 (0) | |

TC, Docetael, and Cyclophosphamide. ddAC-T, dose dense doxorubicin, and cyclophosphamide followed by paclitaxel.

In the group of participants who were overweight or obese at diagnosis, over 80% had higher grade tumors (grade 2 or 3) compared to 72.5% of women in the healthy BMI group. Multifocal tumors were present in 30.2% of participants who were overweight and 25.5% of participants with obesity. Of those with obesity, 76.6% received a lumpectomy, 34.0% received chemotherapy, and 72.3% received radiation. Individuals with obesity were more likely to receive a lumpectomy compared to participants without obesity ($p$ = 0.01). Among all participants, 34.3% received chemotherapy with 40.4% receiving TC, 57.4% receiving ddAC-T, and 2.1% receiving both. About 42.5%, 28.3%, and 34.0% of those with healthy weight/underweight, overweight, and obesity, respectively, received chemotherapy.

At the time of chart review, 7.1% had a recurrence; half had local recurrence, and half had distant recurrence.

### 3.3. BMI Status at Recent Follow-up

The prevalence of obesity among participants increased from 33.6% at diagnosis to 42.1% at most recent follow up ($p$ = 0.012). We tested associations between each initial weight category (underweight, healthy, overweight, obesity, class I obesity, class II obesity, class III obesity, and combined class II + III obesity) with recurrence, metastasis, Nottingham's grade, and stage. No significant association was found. When testing associations between recent BMI status and recurrence, metastasis, Nottingham's grade, and stage, a few significant associations were found. Recent underweight BMI was significantly positively associated with recurrence ($\chi^2(1)$ = 5.62, $p$ = 0.018) and metastases ($\chi^2(1)$ = 10.34, $p$ = 0.001) which may be related to cancer cachexia. Additionally, recent class II obesity alone had a significant association with overall stage of cancer ($\chi^2(2)$ = 8.81, $p$ = 0.012). Black women were statistically more likely to have class II obesity than other BMI statuses at recent follow-up ($\chi^2(2)$ = 12.08, $p$ = 0.002).

### 3.4. Weight Gain

In the present study, 52.9% ($n$ = 74) of participants experienced an increase in BMI from baseline to chart review (Figure 2). Among these participants, average BMI at diagnosis and at most recent visit significantly differed ($p$ < 0.001). On average, BMI at recent follow up was 2.35 kg/m$^2$ higher than BMI at diagnosis (95% CI [1.85, 2.85]). In the overweight group, there was a significant average difference between BMI at diagnosis and BMI at most recent visit ($p$ < 0.001). On average, recent BMI scores for participants in the overweight group were 1.412 kg/m$^2$ higher at most recent visit compared to diagnosis (95% CI [0.62, 2.20]).

Of all participants, 32.1% experienced a $\geq$5% increase in BMI since diagnosis (Figure 3). BMI increases $\geq$5% were observed in 27.5% of the healthy/underweight group, 47.2% of the overweight group, and 19.1% of the group with obesity. We identified a significant association between a $\geq$5% increase in BMI and BMI status at diagnosis (healthy BMI, overweight, obesity) ($\chi^2(2)$ = 9.521, $p$ = 0.009). In particular, the overweight group was more likely to experience a $\geq$5% increase in BMI ($\chi^2(2)$ = 8.83, $p$ = 0.003).

The prevalence of a $\geq$5% increase observed in participants within 36 months of diagnosis, 36–60 months of diagnosis, and greater than 60 months from diagnosis was 25%, 31.4%, and 36.1%, respectively (Figure 4). There was no statistically significant difference in recently recorded BMI between these three time groups (F (2137) = 0.297, $p$ = 0.743). Similarly, participants did not differ in total change in BMI by timepoint group.

At most recent follow-up, about 45% of participants had a $\geq$0.6 kg/m$^2$ increase in BMI, which was consistent for individuals both less than and greater than 5 years since diagnosis. Among those who were overweight or obese at baseline, a $\geq$0.6 kg/m$^2$ increase in BMI was observed in 54.7% of those who were overweight at baseline and 36.2% of those with obesity, respectively. There was no significant association between experiencing a $\geq$0.6 kg/m$^2$ increase in BMI and BMI status at diagnosis (healthy BMI, overweight, obesity) ($\chi^2(2)$ = 3.603, $p$ = 0.165). However, there was a significant association between experiencing a $\geq$0.6 kg/m$^2$ decrease in BMI and initial BMI status ($\chi^2(2)$ = 10.380, $p$ = 0.006). Compared

to other participants, there was a higher proportion of participants with obesity who went on to experience a $\geq 0.6$ kg/m$^2$ decrease (Figure 4).

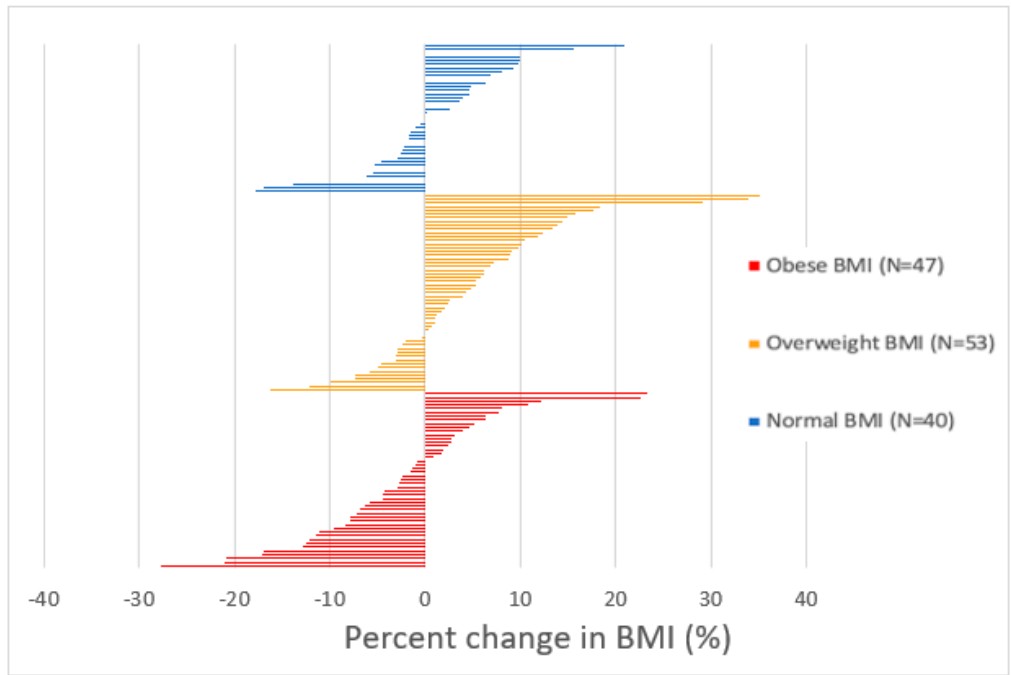

**Figure 2.** Percent change in BMI from baseline to most recent follow up by BMI group. Each bar represents a single participant. The percent change in BMI among all participants ranged from −27.73% to +35.17%. Among those with obesity, percent change in BMI ranged from −27.73% to +23.31%. Among overweight participants, change in BMI ranged from −16.21% to +35.17%. Among the healthy/underweight BMI group, change in BMI ranged from −17.84% to 20.86%.

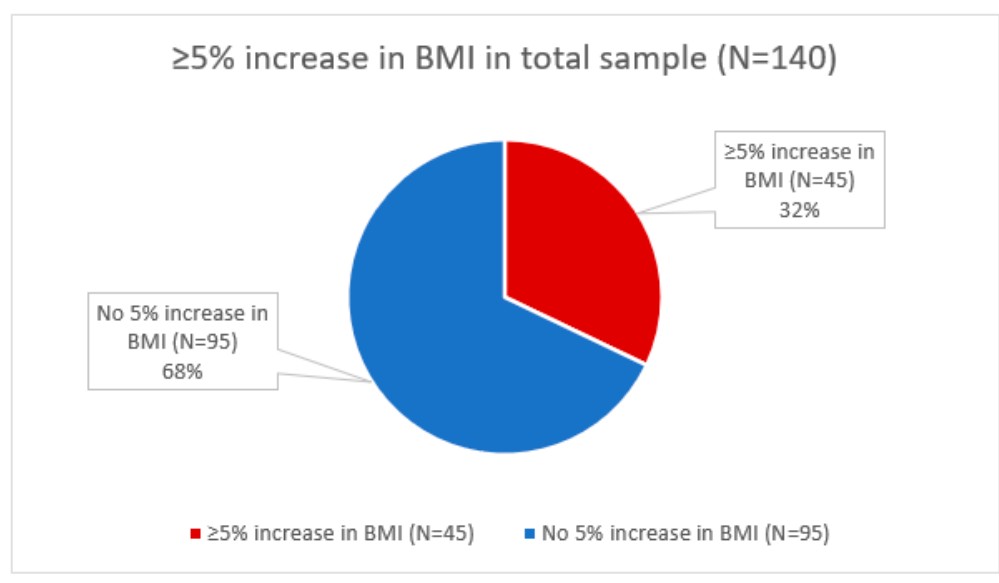

**Figure 3.** Prevalence of at least 5% increase in BMI from date of diagnosis to most recent follow-up.

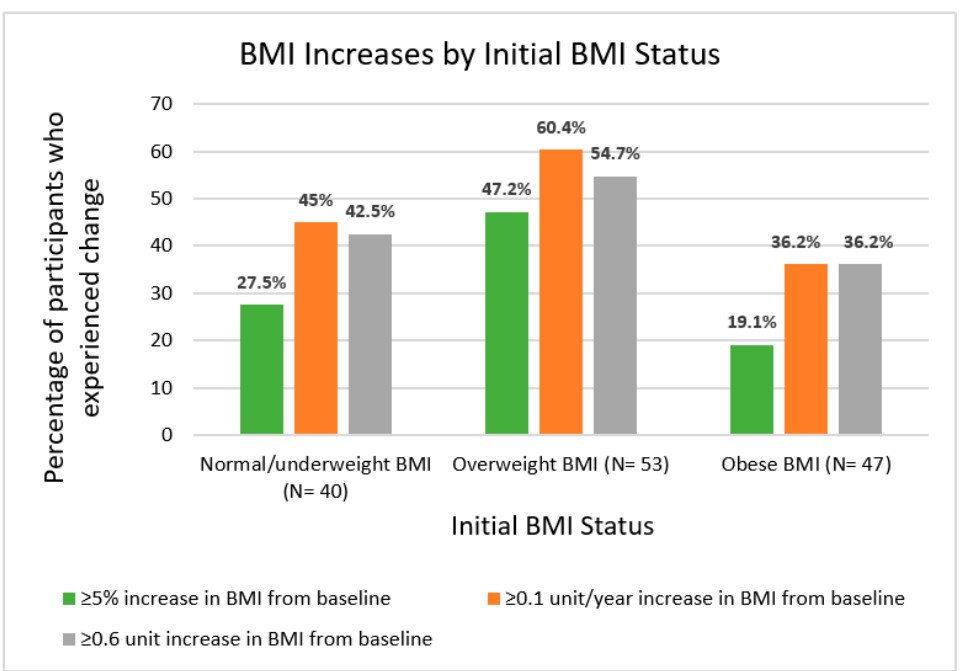

**Figure 4.** Proportion of percent and unit increases in BMI in kg/m² from date of diagnosis to most recent follow-up by BMI group.

Among those within 5 years of diagnosis, 49.4% had a total BMI increase of at least 0.1 per year. Within the healthy BMI, overweight, and obesity categories at baseline, 45.5%, 71.4%, and 31% of participants within 5 years of diagnosis had a total BMI increase of at least 0.1 per year, respectively. Additionally, among those up to 5 years since diagnosis, there was a significant association between experiencing a $\geq$0.1 kg/m² increase per year since diagnosis and baseline BMI status (healthy BMI, overweight, obesity) ($\chi^2(2) = 9.486$, $p = 0.009$). Among women at any timepoint since diagnosis, there were significant differences between initial BMI group and experiencing a $\geq$0.1 kg/m² increase per year. A total of 60.4% of those overweight at baseline experienced a $\geq$0.1 kg/m² increase per year since diagnosis compared with 45% of women within the healthy BMI group ($p = 0.049$). Overweight status at baseline was also associated with a higher risk of experiencing a $\geq$0.1 kg/m² increase per year since diagnosis (OR, 2.264; 95% CI, 1.127, 4.548 [$p = 0.022$]) and any increase in BMI (OR, 3.125; 95% CI, 1.515, 6.448 [$p = 0.002$]). After adjusting for black race, smoking history, current alcohol use, and marital status, the association between overweight status and any BMI increase remained statistically significant (OR, 3.041; 95% CI, 1.462, 6.326 [$p = 0.003$]). There was also no evidence of an interaction between black race and overweight status [$p = 0.901$].

Relationships between weight gain with recurrence and metastasis, grade, and stage were examined. There were no significant relationships identified between these tumor characteristics and outcomes and BMI changes measured ($\geq$5% in BMI, $\geq$0.6 units in BMI, $\geq$0.1 units in BMI/year and decrease of $\geq$0.6 units in BMI, and total change in BMI). However, the relationship between metastasis and a $\geq$5% increase in BMI approached significance ($p = 0.085$).

*3.5. Changes in BMI Status*

From the date of diagnosis to most recent follow-up, 25.7% of participants stayed within the underweight or healthy BMI classification, and 51.4% of participants continued to have obesity or be overweight (Figure 5). While 8.6% shifted from having obesity and being overweight downwards into overweight or healthy BMI categories, 14.3% of participants shifted upwards into the overweight or obesity category. One of the two participants underweight at baseline (50%) shifted to a BMI within the healthy range.

Comparable proportions of black and non-black women with obesity had at least 5% BMI increase from baseline. While half of overweight non-black women had a BMI increase of at least 5% from baseline, only 20% of overweight black women had at least a 5% increase in BMI ($p = 0.201$).

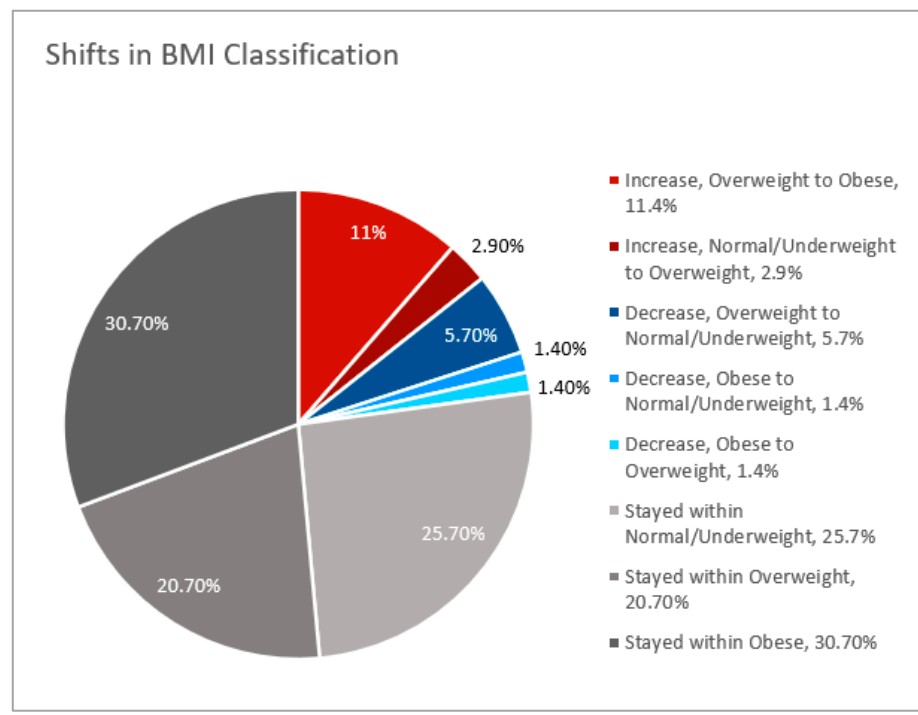

**Figure 5.** Shifts in BMI classification from date of diagnosis to most recent follow-up.

### 3.6. Modelling of Factors Associated with Recurrence

A logistic regression model explored the impact of race and obesity on risk of recurrence, accounting for key factors such as high stage (overall stage above 1), high Nottingham grade (grade above 1), and high pathological size (size above T1). Only black race had a significant effect on recurrence. Additionally, we did not find any significant interaction between race and obesity with regards to odds of recurrence ($p = 0.998$). There also did not appear to be any interaction between race and obesity in predicting these clinical features: overall stage, pathologic tumor size category, Nottingham grade, chemo, radiation, surgery, metastatic disease, recurrence, or multifocality (Table 3).

**Table 3.** Summary of Simple Logistic Regression Analyses for Race, Obesity and Outcome of Recurrence.

| Variables | Model 1 OR [CI] | *p*-Value |
|---|---|---|
| Black | 7.21 (1.48, 35.09) | 0.01 |
| Asian | 3.23 (0.29, 36.48) | 0.34 |
| Obesity | 0.26 (0.04, 1.60) | 0.15 |
| Higher stage | 1.91 (0.36, 10.07) | 0.45 |
| Higher grade | 1.71 (0.18, 16.24) | 0.64 |
| Larger size | 0.84 (0.17, 4.24) | 0.83 |

## 4. Discussion

### 4.1. Summary of Key Findings

The main purpose of our study was to understand the prevalence of excess weight and fluctuation of weight among early-stage breast cancer survivors with HR-positive

tumors over time. We found statistically significant weight gain ($\geq$5% or $\geq$0.1 kg/m$^2$/year increase in BMI) and obesity from diagnosis to most recent follow-up. Additionally, there was a high prevalence of BMI increase observed among those who had an overweight BMI at diagnosis. There was less weight gain in those with class I obesity and an overall decrease in BMI for those with class II obesity. Additionally, black women were more likely to have obesity at diagnosis and obesity was associated with a higher likelihood of receiving a lumpectomy.

### 4.2. Race, Obesity, and Breast Cancer

Associations between race and obesity have been well documented in the literature, with black women having the highest rates and risk of developing obesity compared to other racial groups in the United States [19,20]. Our data supports this, as black women were more likely to have obesity at diagnosis. The greater prevalence of obesity in black women may be attributed to post-partum weight retention, consumption of fast food, limited leisure-time physical activity or neighborhood features. Previous studies demonstrate that, on average, women with BMI in the highest quintile have larger tumor sizes and high mitotic cell count, both being contributing factors to worse breast cancer outcomes [6,21–23]. Additionally, there is a 35–40% increased risk of recurrence and mortality among patients with obesity independent of menopausal status [6,21–23]. As a disproportionate population of black women have obesity, black women may be at greater risk for worse breast cancer outcomes. Interestingly, we found that black women with obesity at diagnosis were more likely to receive a lumpectomy instead of a mastectomy. The factors of obesity and choice of treatment modality should be considered as the disparities in breast cancer outcomes remain prominent between black women and their white counterparts [10,16].

### 4.3. Weight Gain and Obesity

Those with obesity tend to lose weight over time, while those who are overweight tend to gain weight. Our findings show statistically significant changes in BMI and obesity from diagnosis to most recent follow-up, including a $\geq$5% increase or $\geq$0.1 kg/m$^2$ increase in BMI. High frequencies of BMI increase were observed among those who were overweight at diagnosis. Interestingly, there were participants in each group who experienced a $\geq$0.6 kg/m$^2$ decrease, particularly those with obesity at diagnosis. Our data supports a recent cross-sectional analysis of national data in middle-aged and older US adults from 1992–2010, which revealed that the average increase in BMI was largest among those who were overweight [24]. The increase in BMI in individuals who are overweight may reflect changes in obesogenic environments (the food environment, physically inactive lifestyles, and the rise in environmental endocrine disrupting chemicals), which affect certain groups disproportionally. Furthermore, there was less weight gain in those with class I obesity and an overall decrease in BMI for those with class II obesity. Another study also revealed that people with class I or II obesity lost significantly more weight than those with an overweight BMI [25]. Unintentional weight loss may explain these findings, as certain chronic diseases common in individuals with obesity (type 2 diabetes, cardiovascular diseases, and cancer) can lead to weight loss. Additionally, pharmacologic obesity treatment and bariatric surgery is increasingly used to promote weight loss and manage obesity-related comorbidities in individuals with obesity.

### 4.4. Limitations

Obesity is a complex disease involving an excessive amount of body fat, and BMI is often used to diagnose this. However, there may be different metrics that more accurately reflect body fat [10]. In our study, we used BMI as a metric of measuring obesity. However, Kwan and colleagues suggest that while BMI may better reflect a relationship to cancer survival rates among non-Hispanic whites and Hispanics, waist to hip ratio (WHR) may be a better indicator among Asian Americans and African Americans [26]. Since WHR was

not available for our study, it is possible that the results may not be as representative of the extent of adiposity in black women.

Additionally, demographic factors such as tobacco and alcohol use may not be completely captured in chart review. We also did not evaluate social determinants of health or other non-clinical confounding variables (i.e., social support, racism, etc.) that may have contributed to shifts in obesity. Nonetheless, our study does consider extensive clinical characteristics and shifts in BMI over time. Additionally, since the database includes patients from a single institution, overall patterns of care for breast cancer were relatively consistent.

### 4.5. Future Directions

Given the detrimental effects of obesity on breast cancer outcomes, it is critical to highlight the prevalence of excess weight at diagnosis and the urgent need to address weight gain as a component of patient care. Meta-analyses within cancer populations have shown that improvements in cardiorespiratory functioning, reductions in fatigue, and improved metabolic biomarkers may play a positive role in preventing or managing existing breast cancer [27]. A reduction of $\geq 0.6$ kg/m$^2$ has been considered a successful benchmark in reducing adiposity as it correlates with a reduction (a negative upper limit in the 95% prediction interval) of mean percentage of body fat mass [14]. Many studies have evaluated different benchmarks in reducing adiposity (0.1, 0.6, 5%, etc.), but due to the mixed findings there is still more research needed in weight gain after breast cancer diagnosis. Guidelines from the American Cancer Society and the National Comprehensive Cancer Network address the importance of weight management and physical activity for cancer survivors. To address weight management, we have conducted studies to identify scalable and effective weight loss interventions. In the Remote delivery of Practice-based Opportunities for Weight Reduction (POWER-remote) trial, we found that remote coaching (including access to online resources, learning materials, and phone calls) was as effective as in-person coaching.

Adaptive and personalized approaches to weight loss may be necessary to facilitate strategies in breast cancer survivors who are overweight or obese [28]. Our current clinical trials are evaluating the impact of treating underlying insomnia prior to a behavioral weight loss intervention (NCT03542604), and the use of weight loss trajectory as well as the addition of anti-obesity pharmacotherapy to behavioral interventions in individuals unlikely to attain significant weight loss with lifestyle modifications alone (NCT04499950). Ultimately, weight management has the potential to influence non-cancer outcomes through modulation of other diseases that contribute to morbidity in aging women including heart disease, stroke, and diabetes.

Our study highlights the disproportionate burden of obesity and breast cancers. Our study also provides insight into patient populations that may be at higher risk of increases in BMI and development of obesity after breast cancer diagnosis. In particular, women who are initially overweight may be more vulnerable to such disadvantageous weight changes. It is imperative to engage in advocacy and create policy encouraging community resources to prevent and address weight gain among survivors. We hope the findings from our study provide guidance in further understanding patterns of weight gain after a diagnosis of HR-positive breast cancer, and aid in the development of much needed interventions for these patients.

**Author Contributions:** Conceptualization, J.Y.S., A.C.W. and D.S.; Methodology, A.C.W.; Formal Analysis, A.G. and G.E.M.; Data Curation, A.G. and J.Y.S.; Writing—Original Draft Preparation, A.G., G.E.M. and J.Y.S.; Writing—Review & Editing, all authors; Visualization, G.E.M. and J.Y.S.; Supervision, J.Y.S.; Project Administration, A.C.W.; Funding Acquisition, J.Y.S. All authors have read and agreed to the published version of the manuscript.

**Funding:** This research was funded by the National Cancer Institute [R25CA203650].

**Institutional Review Board Statement:** The study was conducted according to the guidelines of the Declaration of Helsinki, and approved by the Institutional Review Board of Johns Hopkins (protocol #NA_00019811 and date of approval 12 January 2008).

**Informed Consent Statement:** Informed consent was obtained from all subjects involved in the study.

**Data Availability Statement:** The data presented in this study are available on request from the corresponding author.

**Conflicts of Interest:** The authors declare no conflict of interest.

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
