# Peer review of "Weight Gain after Hormone Receptor-Positive Breast Cancer"

_curroncol, doi:10.3390/curroncol29060326_

Round 1

Reviewer 1 Report

No comments

Author Response

We appreciate your review of our original research article entitled “Weight Gain after Hormone Receptor-Positive Breast Cancer” on 5/4/22.  We appreciate the favorable review and look forward to sharing our results with the readership of Current Oncology.

Reviewer 2 Report

In the submitted manuscript Goyal et al. investigated obesity in female breast cancer survivors over time, patient and tumor characteristics, treatment and recurrence. They showed that there was a significant weight gain from diagnosis to most recent follow-up, particularly among those who were overweight at diagnosis.

This manuscript is quite well written and study behind it is properly designed. However, there are few things which have to be corrected or further improved.

1) It would be more informative if you have also put obtained actual p-values in 'Abstract'.

2) In both 'Introduction' and 'Materials and methods', the weight designation, i.e., selected BMI categories should be properly cited and stated that you have used international WHO BMI cut-off points (e.g., World Health Organization. (‎2005)‎. Surveillance of chronic disease risk factors : country level data and comparable estimates).

3) Line 57: There should be either "shorter disease-free survival" or "lower disease-free survival rate".

4) Line 109: "TC" and "ddAC-T" abbreviations should be explained, also "OR" and "CI" in both text and tables.

5) Line 133: It is not explained why you used parametric tests like t-test and ANOVA, but non-parameteric Spearman's rank-order correlations, and all that without testing data for normality?!

Also, provide which post-hoc test was used with ANOVA, and whenever you mentioned results of correlation analysis in the manuscript, please provide both correlation coefficient and it's p-value, since correlations should be interpreted according to the extent of the corr. coeff. (e.g., DOI: 10.1213/ANE.0000000000002864 ).

6) Explanation of statistically significant results in tables' footnotes is strange and non-intuitive. In all tables, please provide additional column with actual p-values, both significant and non-sign. You could only additionally highlight those significant by for example font in bold.

7) It is unclear why all of a sudden you decided to present time (line 215) and weight gain (line 255) with median and range or IQR, while in 'Statistical Analysis:" subsection you wrote "Continuous variables, such as age and BMI, were described by mean and standard deviation (SD).".

8) Line 305: Non-significant p-value "p = ns" should be presented as in all other situations in the text.

9) It is unclear what means "t73" (line 237) and "t52" (line 240).

10) It is quite reasonable that authors omitted cases of male breast cancers (MBC) since they had only 4 cases (Figure 1), but since obesity is a well known risk factor for MBC, in my opinion this should be also mentioned in 'Introduction'.

11) Since "person is considered to be a survivor from the time of diagnosis until the end of life" statement "more than 3.8 million breast cancer survivors in the United States" (lines 28-29) is insufficiently precise. Is it at the moment, yearly or something else?

Reviewer 3 Report

Goyal et al. investigated the correlation between weight gain and other disease statistics after hormone receptor-positive breast cancer treatment. Below you can find my line-by-line comments for this paper. 

Major comments:

Line 229-231: Your result regarding the correlation between current underweight BMI and metastasis might be due to cancer cachexia. Since this is an important point to highlight, you can highlight this point with a sentence.

In the discussion part, you first summarized your results and tried to support your data with recent studies, however, you also need to add a few sentences to address (the) reason/s behind seeing these results. For example, what could be the possible reason of seeing overweight patients to gain weight, while current obese patients were losing weight over the course of the treatment. Like in this example, it is better to support each findings with your comments and possible explanations. 

Minor comments:

- Line 2: The title needs to be detailed in terms of which stage of the disease is targeted in the study. I think it will be more clear to change the title to 'The effect of weight gain during hormone receptor-positive breast cancer: From diagnosis to treatment'.  

- Line 54-55: There is no need to write Black/White identities starting with an uppercase letter. 

- In Table 1, fonts should be organized more carefully.

- Line 213: Acronyms should be written in full format when they are used for the first time.
